# The Role of m^6^A Modification and m^6^A Regulators in Esophageal Cancer

**DOI:** 10.3390/cancers14205139

**Published:** 2022-10-20

**Authors:** Yuekao Li, Chaoxu Niu, Na Wang, Xi Huang, Shiru Cao, Saijin Cui, Tianyu Chen, Xiangran Huo, Rongmiao Zhou

**Affiliations:** 1Department of Computed Tomography, The Fourth Hospital of Hebei Medical University, Shijiazhuang 050011, China; 2Department of Surgery, Shijiazhuang Ping’an Hospital, Shijiazhuang 050021, China; 3Hebei Provincial Cancer Institute, The Fourth Hospital of Hebei Medical University, Shijiazhuang 050011, China

**Keywords:** m^6^A modification, m^6^A regulators, occurrence, progression, treatment, prognosis, esophageal cancer

## Abstract

**Simple Summary:**

N^6^-methyladenosine (m^6^A) modification and m^6^A regulators play important roles in the occurrence and development of various cancers. Esophageal cancer (ESCA), one of the most common gastrointestinal tumors, seriously affects people’s health. In this review article, we summarized the role and possible mechanism of m^6^A modification and m^6^A regulators in the occurrence, progression, remedy and prognosis of ESCA, which might be useful for determining the pathogenesis of ESCA and providing direction for the future research of ESCA.

**Abstract:**

N^6^-methyladenosine (m^6^A) modification, the most prevalent RNA modification, is involved in all aspects of RNA metabolism, including RNA processing, nuclear export, stability, translation and degradation. Therefore, m^6^A modification can participate in various physiological functions, such as tissue development, heat shock response, DNA damage response, circadian clock control and even in carcinogenesis through regulating the expression or structure of the gene. The deposition, removal and recognition of m^6^A are carried out by methyltransferases, demethylases and m^6^A RNA binding proteins, respectively. Aberrant m^6^A modification and the dysregulation of m^6^A regulators play critical roles in the occurrence and development of various cancers. The pathogenesis of esophageal cancer (ESCA) remains unclear and the five-year survival rate of advanced ESCA patients is still dismal. Here, we systematically reviewed the recent studies of m^6^A modification and m^6^A regulators in ESCA and comprehensively analyzed the role and possible mechanism of m^6^A modification and m^6^A regulators in the occurrence, progression, remedy and prognosis of ESCA. Defining the effect of m^6^A modification and m^6^A regulators in ESCA might be helpful for determining the pathogenesis of ESCA and providing some ideas for an early diagnosis, individualized treatment and improved prognosis of ESCA patients.

## 1. Introduction

N^6^-methyladenosine (m^6^A) refers to the methylation at the N^6^ position of adenosine, which is considered to be the most prevalent RNA modification. Approximately one to two m^6^A residues are found in every 1000 nucleotides [1,2], and are mainly located in the RRACH sequence (R = A or G, H = A, C, or U) [3,4]. However, not all RRACH motifs are methylated, which suggests that methylation at the N^6^ position of adenosine is specific and selective [5,6]. N^6^-methyladenosine occurs in mRNA, rRNA, long non-coding RNA (lncRNA), microRNA (miRNA), circular RNA (circRNA), etc., and is involved in all aspects of RNA metabolism, including RNA processing, nuclear export, stability, translation and degradation. Therefore, m^6^A modification can participate in various physiological functions, such as tissue development, heat shock response, DNA damage response, circadian clock control and even in carcinogenesis through regulating the expression or structure of the gene [5,6,7,8,9,10,11,12,13,14,15,16,17,18,19].

Esophageal cancer (ESCA) is one of the most common malignant tumors, ranking tenth all over the world in 2020 [20]. Esophageal squamous cell carcinoma (ESCC) and esophageal adenocarcinoma (EAC) are the main pathological subtypes of ESCA. Genetic factors and environmental factors interplay to cause the occurrence of ESCA. Smoking, drinking, obesity, etc., are associated with a higher risk of ESCA. Genome-wide association studies and candidate gene association studies both found some susceptible genes of ESCA [21,22,23,24,25]. Furthermore, epigenetic modifications, such as DNA methylation, RNA methylation, histone modifications, et al., also play a critical role in the occurrence and development of ESCA [26,27]. The m^6^A modification is the most frequent RNA modification, and was firstly identified in 1974 [28]. It was considered as a static process until the fat mass and obesity-associated (FTO) gene was firstly found to be a demethylase in 2010 [29]. With the discovery of FTO and the application of m^6^A detection technology, m^6^A modification became a hotspot in the research field of cancer, including ESCA. Recent studies indicated that aberrant m^6^A modification and the dysregulation of m^6^A regulators, the proteins that participate in the m^6^A-modification-related process, played critical roles in the initiation and progression of various cancers. At present, the pathogenesis of ESCA remains unclear. Owing to a poor early diagnosis rate, ESCA patients are commonly diagnosed at advanced stages. Although great progress has been made in the treatment of esophageal cancer, the five-year survival rate of advanced ESCA patients is still dismal because of uncontrolled tumor growth and relapse. Therefore, we systematically reviewed the recent studies of m^6^A modification and m^6^A regulators in ESCA and comprehensively analyzed the role and possible mechanism of m^6^A modification and m^6^A regulators in the occurrence, progression, remedy and prognosis of ESCA. Defining the effect of m^6^A modification and m^6^A regulators in ESCA might be helpful for determining the pathogenesis of ESCA and providing some ideas for an early diagnosis, individualized treatment and improved prognosis of ESCA patients.

## 2. m^6^A Modification and m^6^A Regulators

N^6^-methyladenosine modification is a dynamic and reversible process. In this process, methyltransferases are responsible for the addition of methyl, while removals of m^6^A are performed by demethylases. The fate of m^6^A-modified RNA depends on the protein that recognizes and binds to it. All of the methyltransferases, demethylases and m^6^A-modified RNA bound proteins that take part in the m^6^A-modification-associated process belong to m^6^A regulators.

### 2.1. Methyltransferases

Methyltransferases are also called “writers”. They catalyze the installation of m^6^A in the form of a complex, the m^6^A methyltransferase complex (MTC), which consists of methyltransferase-like 3 (METTL3), methyltransferase-like 14 (METTL14), Wilms = tumor-1-associated protein (WTAP), RNA binding protein 15 (RBM15), RNA binding protein 15B (RBM15B), vir like m^6^Amethyltransferase associated (VIRMA) (also known as KIAA1429), zinc finger CCCH-type containing13 (ZC3H13) and HAKAI. METTL3, METTL14 and WTAP constitute the catalytic core of MTC. METTL3 is the main subunit of MTC, with catalytic activity, and can bind S-adenosylmethionine (SAM) and transfer methyl to the N^6^ position of adenosine [30]. In addition, cytoplasmic METTL3 can bind to the m^6^A-modified 3′-untranslated region (UTR) of target mRNA, recruit eukaryotic translation initiation factor 3 subunit H (EIF3H) and enhance the translation efficiency [31,32]. METTL14 forms a stable heterodimer with METTL3 at a 1:1 ratio, stabilizes METTL3 and recognizes and binds to target RNA through its C-terminal RGG repeats [30,33]. With the help of WTAP, the METTL3-METTL14 heterodimer can be located in the nuclear speckle [34]. In addition to the catalytic core of MTC, other components are associated with WTAP and are helpful for the recruitment and localization of the METTL3-METTL14 heterodimer [35,36,37,38,39]. In a WTAP-dependent manner, RBM15/RBM15B correlates with METTL3 and recruits MTC to U-rich regions immediately adjacent to RRACH motifs to catalyze the deposition of m^6^A [37,40,41]. VIRMA is also a WTAP-associated factor and recruits MTC to 3′-UTR and near the stop codon for m^6^A catalysis [39]. ZC3H13 connects RBM15/RBM15B with WTAP through its C-terminal structure domain [36,38]. The knockdown of ZC3H13 leads to translocation from the nucleus to cytoplasm for a large proportion of WTAP, VIRMA, HAKAI, METTL3 and METTL14, indicating the important role of ZC3H13 for the nuclear localization of MTC [38]. HAKAI is an E3 ubiquitin-protein ligase, while its role in m^6^A catalysis is unclear. It is worth noting that m^6^A deposition depends on transcription. METTL3 and METTL14 form a heterodimer in the cytoplasm and then the heterodimer enters the nucleus with the help of a nuclear localization signal in METTL3. METTL4 can recognize and bind to histone H3 trimethylation at Lys36 (H3k36me3), promote the binding of MTC with RNA polymerase II and transfer MTC to actively transcribing RNAs to install m^6^A cotranscriptionally [42].

In addition to MTC, there are other methyltransferases, such as methyltransferase-like protein 16 (METTL16), cap-specific adenosine methyltransferase (CAPAM), methyltransferase-like protein 5 (METTL5)/tRNA methyltransferase activator subunit 11-2 (TRMT112) complex and zinc finger CCHC-type containing 4 (ZCCHC4). METTL16 catalyzes m^6^A deposition in the A43 of U6 small nuclear RNA (snRNA), and is involved in the splicing of RNA [43,44]. The m^6^A 43 is considered to affect the interaction between snRNA and pre-mRNA and thus regulate the splicing of pre-mRNA [44]. Methionine adenosyl transferase 2A (MAT2A) encodes SAM synthetase. The 3′-UTR hairpins of MAT2A mRNA are substrates of METTL16. The m^6^A modification of 3′-UTR hairpins influences the splicing of MAT2A pre-mRNA and maintains SAM homeostasis [43,45]. If 2′-O-methyladenosine (Am) is the first transcribed nucleotide of eukaryotic capped mRNAs, CAPAM can recognize it and deposit m^6^A on it to form a m^7^GPPPm^6^Am motif [46,47]. The METTL5/TRMT112 complex and ZCCHC4 are responsible for the methylation of the A1832 of 18S and A4220 of 28S rRNA, respectively [48,49,50].

### 2.2. Demethylases

Demethylases are termed as “erasers”. The m^6^A demethylation occurs on nascent transcripts. FTO is the first identified demethylase, and alkB homolog 5 (ALKBH5) is the second one. Ferrous iron and α-ketoglutarate are cofactors of FTO and ALKBH5 [51]. Both FTO and ALKBH5 can remove m^6^A modification on single RNA and DNA [8,52]. In addition, FTO also can act as demethylase for N^6^,-2′-O-dimethyladenosine (m^6^Am) near the N^7^-methylguanosine (m^7^G) cap [53].

### 2.3. m^6^A RNA Binding Proteins

The fate of m^6^A-modified RNA depends on the protein that binds to it. This kind of RNA binding protein is referred to as a “reader”. Readers include YT521-B homology (YTH) family proteins, insulin like growth factor 2 mRNA binding proteins (IGF2BPs), heterogeneous nuclear ribonucleoproteins (HNRNPs) and eukaryotic translation initiation factor 3 (EIF3).

## 3. m^6^A Modification and Its Effect on Various RNAs in ESCA

A proper m^6^A level, mainly relying on the appropriate expression and function of m^6^A regulators, is necessary for sustaining normal bioprocesses. The disruption of the dynamic balance between the installation and removal of m^6^A modification will lead to the development of diseases, including cancer. An aberrant m^6^A level associated with the dysregulation of m^6^A regulators has been reported in a variety of cancers, such as gastric cancer, hepatocellular carcinoma, bladder cancer, etc. [54,55,56]. Nucleotide sequence changes could also result in the gain or loss of m^6^A sites and contribute to carcinogenesis [57,58]. For example, a base transition from the G to A of rs5746136 in SOD2 led to an increased m^6^A modification level of SOD2 and an increased binding of HNRNPC with SOD2 through an “m^6^A switch” mechanism followed by the upregulation of SOD2. The overexpression of SOD2 inhibited the proliferation, migration and invasion of bladder cancer cells, which suggested that SOD2 acted as a tumor suppressor gene for bladder cancer. Thus, the A allele of rs5746136 in SOD2 was associated with a reduced risk of bladder cancer [57]. m^6^A could deposit on various type of RNA, such as coding RNA and non-coding RNA, participate in all steps of RNA metabolism and post-transcriptionally regulate the expression of the gene (Table 1) [59,60,61,62,63,64,65,66,67,68,69,70,71,72,73,74,75,76,77,78,79,80,81,82,83].

### 3.1. m^6^A Modification in ESCA

Most studies revealed that the m^6^A level was elevated in ESCC tissues and cell lines [73,84,85,86]. However, Cui et al. [83] found lower m^6^A levels in ESCC cell lines and tissues compared to their counterparts. Some studies indicated that the m^6^A level could act as a diagnostic and prognostic marker. In gastric cancer, the m^6^A level might be used to distinguish patients from healthy individuals and predict the prognosis of patients treated with immunotherapy [87,88]. In lung cancer, a higher m^6^A level in circulating tumor cells than in whole blood cells might be associated with tumor metastasis [89]. Inconsistent results of the m^6^A level in ESCC suggested that further large sample studies are needed to draw a reliable conclusion.

### 3.2. The Effect of m^6^A Modification on mRNA in ESCA

The fate of m^6^A-modified RNA relies on the m^6^A reader that binds to it. YTH family proteins, consisting of YTH m^6^A-binding protein 1 (YTHDF1), YTHDF2, YTHDF3, YTH domain-containing 1 (YTHDC1) and YTHDC2, contain a specific YTH domain, through which, they can recognize and bind to target RNA in an m^6^A-dependent way [90,91]. YTHDF1 promotes translation initiation and protein synthesis [10]. On the contrary, YTHDF2, the first discovered m^6^A reader, enhances the degradation of m^6^A-modified mRNA, either by delivering them to the mRNA decay site or recruiting the CCR4-NOT deadenylase complex to initiate mRNA degradation [9,92]. YTHDF3 interacts with YTHDF1 or YTHDF2, playing opposite roles by promoting mRNA translation or enhancing mRNA degradation [93,94]. YTHDC1 regulates the splicing of the exon and promotes the translocation of m^6^A-modified mRNA from the nucleus to the cytoplasm [95,96,97]. YTHDC2 elevates the translation efficiency of m^6^A-modified mRNA; accordingly, the abundance of target mRNA is reduced [98,99]. IGF2BPs, including IGF2BP1, IGF2BP2 and IGF2BP3, enhance mRNA stability [100]. HNRNPs contain HNRNPA2B1, HNRNPC and HNRNPG. HNRNPC and HNRNPG regulate the alternative splicing of mRNA in an m^6^A-dependent way [13,101]. EIF3 can act as reader of m^6^A in the 5′-UTR of mRNA [17]. EIF3 participates in almost all steps of translation initiation, which is a rate-limiting process. EIF3 promotes the formation of the 43S pre-initiation complex (PIC), bridges 43S PIC and mRNA bound to the EIF4F complex and takes part in the AUG start codon scanning process [102,103,104,105].

It is generally acknowledged that YTHDF1 plays a facilitating role in translation initiation and protein synthesis [10]. That was true in the study from Zhao et al. [73], which demonstrated that YTHDF1 upregulated the protein level of ERBB2 through recognizing m^6^A-modified ERBB2 mRNA. Conversely, the knockdown of YTHDF1 enhanced the protein level of HSD17B11, which suggested that YTHDF1 decreased the translation efficiency of m^6^A-modified HSD17B11 mRNA [74]. YTHDF2 promoted the degradation of APC mRNA and decreased APC expression via binding to m^6^A-modified APC mRNA [69]. IGF2BP2 enhanced the stability of m^6^A-modified TK1 and KIF18A mRNA and upregulated their expression [66]. In addition, m^6^A readers could regulate the stability and expression of their downstream mRNAs through interacting with lncRNA [63,64,65,68,70]. For example, LBX2-AS1 and HNRNPC synergized to increase the stability of ZEB1 and ZEB2 mRNA, upregulated their expression and consequently promoted the migration and epithelial mesenchymal transition (EMT) of ESCC cells [68].

### 3.3. The Effect of m^6^A Modification on Non-Coding RNAs in ESCA

Although without a coding ability, non-coding RNAs (ncRNAs), such as miRNA, lncRNA and circRNA, serve a critical role in regulating the expression of the gene.

The miRNAs bind to the 3′-UTR of the target mRNA, and then silence or inhibit the expression of corresponding genes. miRNA biogenesis includes the following steps: firstly, primary miRNA (pri-miRNA) is transcribed from DNA; secondly, pri-miRNA is cleaved into precursor miRNA (pre-miRNA), which requires a microprocessor complex composed of drosha ribonuclease III (DROSHA) and DiGeorge syndrome critical region 8 (DGCR8); thirdly, pre-miRNA is cleaved into mature miRNA. In ESCC, m^6^A writers and erasers deposit and remove m^6^A on pri-miRNA, respectively [75,77,78,79]. For instance, METTL3 elevates the m^6^A level of pri-miR-200-5p, whereas ALKBH5 reduces the m^6^A level of pri-miR-194-2 [75,78]. A previous study showed that HNRNPA2B1 played a crucial role in the maturation of miRNA through recognizing m^6^A on pri-miRNA and interacting with DROSHA and DDGCR8 [106]. In ESCC, HNRNPA2B1 promoted the proliferation of ESCC cells through binding to a m^6^A-modified miR-17-92 cluster and upregulating the expression of a miR-17-92 cluster [79].

m^6^A modification could also be found on lncRNAs, which might be involved in the regulation of gene expression via influencing the interaction of lncRNAs with RNA binding proteins through an “m^6^A switch” mechanism or impacting the interaction between lncRNAs and miRNAs [13,107]. In ESCC, the overexpression of FTO significantly reduced the enrichment of m^6^A at site 2 of the LINC00022 transcript and led to a decrease in the degradation of LINC00022 by YTHDF2 [83]. The study of Wu et al. [81] showed that lncRNA LINC00278 encoded a micropeptide named Yin Yang 1 (YY1)-binding micropeptide (YY1BM), and the binding of YTHDF1 to m^6^A-modified LINC00278 led to an increased translation of YY1BM. LncRNA metastasis-associated lung adenocarcinoma transcript 1 (MALAT1) is located at nuclear speckles (NSs). The MALAT1-m^6^A-enriched sequence and the binding of YTHDC1 to the m6A of MALAT1 were necessary for maintaining the composition of NSs and migratory capability of ESCC cells [82]. In addition, as mentioned above, lncRNA could interact with m^6^A readers to regulate the expression of target mRNAs [64,65,68,70].

circRNAs perform different biological functions based on their diverse distribution in cells. Nuclear circRNAs might affect transcription and splicing [108,109]. Cytoplasmic circRNAs might not only absorb miRNAs and alleviate their depression of the target mRNA [110,111] but also might interact with RNA binding proteins and enhance their functional impacts [112,113]. It is worth mentioning that the exon-derived circRNAs might have a protein-encoding ability [114,115]. m^6^A regulators regulated the expression, distribution and function of circRNAs through installing, removing and recognizing m^6^A on circRNAs in all sorts of cancers [112,113,116,117,118,119]. In gastric cancer and cervical cancer, METTL14 and ALKBH5 acted as a transmethylase of circORC5 and demethyltransferase of circCCDC134, respectively [116,117]. YTHDC1 facilitated m^6^A-modified circMET and circNSUN2 exportation from the nucleus to cytoplasm in NONO-TFE3 fusion renal cell carcinoma and colorectal carcinoma, respectively [112,113]. IGF2BP2 interacted with circNSUN2 to increase the stability of HMGA2 mRNA and promote the metastasis of colorectal carcinoma [113]. IGF2BP1 promoted the translation of m^6^A-modified circMAP3K4 into circMAP3K4-455aa via recognizing m^6^A modification on circMAP3K4 in hepatocellular carcinoma [118]. Interestingly, m^6^A modification on circALG1 enhanced its ability as competitive endogenous RNA (ceRNA) of miR-342-5p by increasing its binding to miR-342-5p in colorectal cancer [119]. It is a pity that no study has reported how m^6^A modification regulates the expression and function of circRNAs during the development and progression of ESCA until now, which provides a direction for future research in ESCA.

## 4. The Role of m^6^A Regulators in Development, Progression, Prognosis and Treatment of ESCA

ESCC and EAC are the main pathological subtypes of ESCA. To date, only two studies have been conducted to investigate the expression of m^6^A regulators and their association with clinicopathological characteristics and prognosis in EAC. In some of the studies, although the histological type of tissue samples employed was not indicated clearly, the cells used for cytology experiments were ESCC cells. Therefore, these studies were described alongside studies about ESCC.

### 4.1. m^6^A Regulators and EAC

Plum et al. [120] examined the expression of IGF2BP3 in 371 EAC samples, including 109 early invasive EAC (pT1a/pT1b) and 262 locally advanced EAC (>pT2), with a higher IGF2BP3 expression in locally advanced EAC. The pT1a and pT1b were used to describe a cancerous lesion restricted to the mucosa and submucosa, respectively. The pT1b was further divided into sm1, sm2 and sm3. The IGF2BP3 expression had an elevated trend with an increase in the invasive depth from pT1a to pT1b (sm3). A high IGF2BP3 expression predicted a shorter survival for early invasive EAC patients. The researchers thought that IGF2BP3 might be useful for therapeutic decisions in early invasive EAC. In addition, Burdelski et al. [121] reported an association of a high IGF2BP3 expression with adverse features such as a high grade and metastatic phenotype in EAC.

### 4.2. m^6^A Regulators and ESCA (Not including EAC)

#### 4.2.1. The Expression of m^6^A Regulators and their Association with Clinicopathological Characteristic in ESCA

The expression, clinical significance and biological function of m^6^A regulators in ESCA are shown in Appendix A [59,60,61,66,69,72,73,74,75,76,79,80,83,84,85,86,122,123,124,125,126,127,128,129,130,131,132,133,134,135,136,137,138,139].

Writers

Methyltransferases, including METTL3, METTL14, WTAP, RBM15 and KIAA1429, were upregulated in ESCA. In addition, METTL3 expression was associated with sex, tumor size, depth of invasion, differentiation extent, lymph node metastasis, distant metastasis and TNM stage [59,72,75,85,123,125,126,127,128]. WTAP expression was associated with TNM stage and lymph node metastasis [129].

Erasers

The expression of ALKBH5 was downregulated in ESCA tissues [80,86,122,130,131], which was more frequently observed in advanced ESCC such as T3–T4, N1–N3, clinical stage III–IV and histological grade III tumors [122,130,131]. FTO was reported to be increased in ESCA tissues and cell lines in most studies [73,74,83,132]. Female patients and advanced stage patients had a significantly higher FTO expression than male patients and early stage patients, respectively [83]. On the contrary, Zhang et al. [122] found a lower FTO expression in ESCA tissues.

Readers

A higher expression of YTHDF1, IGF2BP1, IGF2BP2, HNRNPA2B1, HNRNPC, EIF3B, EIF3E and EIF3H was found in ESCA [66,79,84,127,130,134,135,136,137,138,139]. Moreover, there was a positive correlation of HNRNPA2B1 expression with tumor diameter, lymphatic metastasis, distant metastasis and advanced lymph node stage [79,84]. A significant discrepancy of EIF3B expression was shown in different invasion depths, with or without lymph node metastasis and different TNM stages [137]. Takata et al. [140] tested the expression of IGF2BP3 in 191 ESCC specimens using an IHC assay and found 113 tumors that were IGF2BP3-positive. The expression of IGF2BP3 was related to the depth of invasion, lymph node metastasis and pathological stage. The expression of YTHDF2 and YTHDC2 was downregulated in ESCA [83,133].

The expression of m^6^A regulators was dysregulated in ESCA and was associated with a clinicopathological characteristic of ESCA, which implied an important role of m^6^A regulators in the development and progression of ESCA. To date, all studies about METTL3 in ESCA showed an upregulation of METTL3. Therefore, METTL3 expression had the potential as a diagnostic marker for ESCA, with a sensitivity of 75% and specificity of 72.06% [85]. In addition, a high standard uptake value (SUVmax) of ^18^F-FDG PET/CT could be used as a non-invasive predictive marker for a high METTL3 expression of ESCA patients [125]. The expression of ALKBH5 was downregulated in ESCA. The difference in sample size and detection methods might contribute to inconsistent results about FTO. The amount of studies on other m^6^A regulators is insufficient to draw a preliminary conclusion. The alteration of the nucleotide sequence and epigenetic modification could both impact the expression of m^6^A regulators. For instance, rs2416282 regulated YTHDC2 expression [133]. KAT2A and SIRT2 activated and inhibited the transcription of METTL3 through increasing and decreasing the H3K27 acetylation level in the METTL3 promoter region, respectively [72]. m^6^A regulators not only regulated the expression of various type of RNAs but also received regulation by non-coding RNAs (Table 2) [66,80,122,129,135,136]. LncRNA EMS sponged miR-758-3p, weakened its repression of WTAP and led to an increased expression of WTAP [129]. Furthermore, external stimulus could change the expression of m^6^A regulators. Specifically, cigarette smoking decreased the methylation of the ALKBH5 promoter, enhanced ALKBH5 expression and affected the progression of male ESCC [81].

#### 4.2.2. The Effect and Mechanism of m^6^A Regulators in Progression of ESCA

The m^6^A regulators affected the biological behavior of ESCC cells through the same or a different gene/signaling pathway, which is shown in Figure 1, Figure 2, Figure 3, Figure 4, Figure 5, Figure 6 and Figure 7 [59,60,61,62,63,64,65,66,68,69,70,71,72,73,74,75,76,78,79,80,81,82,83,84,85,86,124,126,127,131,132,133,134,135,136,137,138,139,141].

##### m^6^A Regulators Regulated Biological Behavior of ESCC Cells in an m^6^A-Dependent Way

Writers

METTL3 could promote the proliferation, migration and invasion of ESCC cells and inhibited ESCC cells’ apoptosis in an m^6^A-dependent way [59,60,61,75,76,85,122]. As methyltransferase, METTL3 affected the expression of coding RNA and the biogenesis of microRNA through changing the m^6^A level. In METTL3 knockdown cells, m^6^A peaks in the UTRs of glutaminase 2 (GLS2) and expression of GLS2 were decreased, which suggested that METTL3 could upregulate GLS2 expression through increasing the m^6^A level of GLS2. GLS2 knockdown alleviated the migration and invasion ability of ESCC cells increased by METTL3 overexpression. That is, METTL3 could regulate GLS2 expression in an m^6^A-dependent way and GLS2 mediated the effect of METTL3 on the malignant phenotype of ESCC cells [59]. Similarly, METTL3 played a crucial role in ESCC initiation and progression through regulating the expression of NOTCH1 in an m^6^A-dependent manner and activating the Notch signaling pathway [60]. The downregulation of METTL3 decreased the m^6^A level of pri-miR-200-5p, inhibited the binding of DGCR8 with pri-miR-200-5p and reduced the expression of miR-200-5p. The upregulation of miR-200-5p could target and inhibit nuclear factor IC (NFIC), which had been reported to play a suppressive role in the proliferation, metastasis and EMT process of ESCC cells [142]. Therefore, METTL3 influenced the malignancy of ESCC cells through regulating the expression of miR-200-5p and NFIC [75]. Similarly, the overexpression of METTL3 elevated the expression of miR-320b, which inhibited the expression of programmed cell death 4 (PDCD4), and then activated the AKT signaling pathway and led to lymphangiogenesis and the lymphatic metastasis of ESCC [76] (Figure 1).

**Figure 1 cancers-14-05139-f001:**
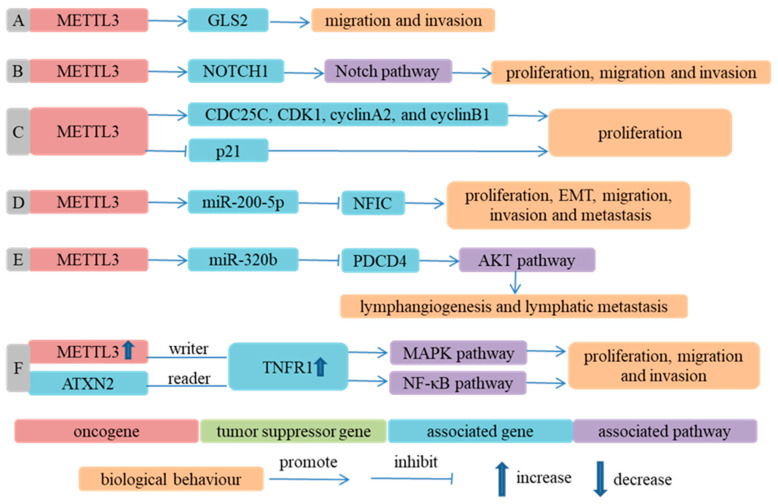
Methyltransferases participate in ESCA progression in an m^6^A-dependent way. (**A**) Ref. [59] METTL3 upregulated GLS2 expression through increasing m^6^A modification level of GLS2 and promoted migration and invasion of ESCC cells; (**B**) Ref. [60] METTL3 upregulated NOTCH1 expression through increasing m^6^A modification level of NOTCH1, activated Notch signaling pathway and thus promoted proliferation, migration and invasion of ESCC cells; (**C**) Ref. [85] Depletion of METTL3 significantly reduced m^6^A modification levels of ESCC cells. METTL3 promoted proliferation of ESCC cells by increasing CDC25C, CDK1, cyclin A2 and cyclin B1 amounts and reducing p21 amounts; (**D**) Ref. [75] METTL3 upregulated miR-200-5p expression through increasing m^6^A modification level of pri-miR-200-5p, inhibited expression of NFIC and promoted malignant biological behavior of ESCC cells; (**E**) Ref. [76] METTL3 elevated miR-320b expression through increasing m^6^A modification level of pri-miR-320b, inhibited expression of PDCD4 and then activated AKT signaling pathway, leading to lymphangiogenesis and lymphatic metastasis of ESCC cells; (**F**) Ref. [61] ATXN2 bound to TNFR1 methylated by METTL3 (upregulated in ESCC samples) and augmented translation of TNFR1. TNFR1 promoted malignant behavior of ESCC cells, such as proliferation, migration and invasion, by activating MAPK and NF-κB signaling pathways.

Erasers

Whether ALKBH5 promoted or inhibited the progression of ESCA was uncertain. ALKBH5 inhibited the biogenesis of miR-194-2 through decreasing its m^6^A modification, upregulated the expression of RAI1, activated the Hippo signaling pathway and suppressed the growth and motility of ESCC cells in vitro and in vivo [78]. Conversely, Nagaki et al. [62] reported that the depletion of ALKBH5 suppressed the proliferation and migration of ESCC cells. Mechanistically, ALKBH5 knockdown increased the m^6^A modification and stability of CDKN1A mRNA and upregulated the expression of p21, a cell cycle inhibitor. Furthermore, the knockdown of ALKBH5 suppressed tumor growth in nude mice. Collectively, ALKBH5 downregulated the expression of p21 in an m^6^A-dependent way and facilitated cell cycle progression and the proliferation of ESCC cells in vitro and in vivo. LncRNAs may influence mRNA expression through m^6^A RNA modification [143,144]. Qin et al. [63] reported that cancer susceptibility candidate 15 (CASC15) interacted with FTO, reduced the m^6^A level of single-minded 2 (SIM2) mRNA, decreased SIM2 expression and facilitated ESCC progression (Figure 2).

**Figure 2 cancers-14-05139-f002:**
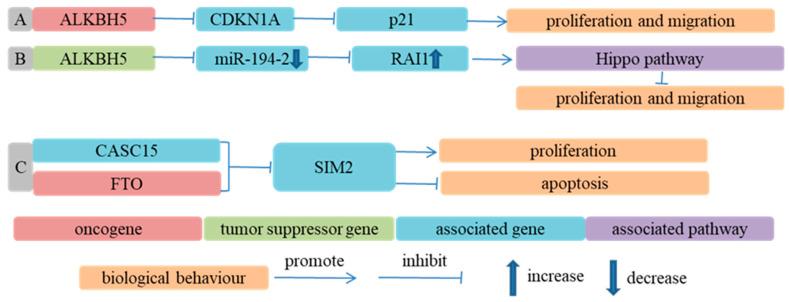
Demethylases participate in ESCA progression in an m^6^A-dependent way. (**A**) Ref. [62] ALKBH5 downregulated expression of p21 by decreasing m^6^A modification level of CDKN1A and promoted proliferation and migration of ESCC cells; (**B**) Ref. [78] ALKBH5 inhibited biogenesis of miR-194-2 through decreasing its m^6^A modification level, upregulated expression of RAI1, activated Hippo signaling pathway and suppressed malignant biological behavior of ESCC cells; (**C**) Ref. [63] FTO, as demethyltransferase of SIM2, interacted with CASC15 to decrease SIM2 expression and facilitate progression of ESCC.

Readers

IGF2BP2 could increase the stability of mRNA individually [66] or in combination with lncRNA [64,65]. LncRNA colon-cancer-associated transcript 2 (CCAT2) absorbed miR-200b, weakened the inhibition of miR-200b on IGF2BP2 and upregulated the expression of IGF2BP2. The overexpression of IGF2BP2 enhanced the expression of TK1 by recognizing the m^6^A of TK1 mRNA, which promoted the oncogenesis of ESCC cells in nude mice and facilitated the migration and invasion of ESCC cells [66]. The interaction of IGF2BP2 with lncRNA small nucleolar RNA host gene 12 (SNHG12) and the interaction of IGF2BP2 with lncRNA forkhead box P4 antisense RNA 1 (FOXP4-AS1) could enhance the mRNA stability of CTNNB1 and forkhead box P4 (FOXP4), respectively, and partly contributed to the proliferation, migration, EMT and stemness of ESCC cells [64,65].

HNRNPA2B1 could increase the expression of the miR-17-92 cluster in an m^6^A-dependent manner, promote the proliferation of ESCC cells and affect the prognosis of ESCA patients negatively [79]. Similarly, HNRNPA2B1 might also facilitate ESCC progression by accelerating fatty acid synthesis via increasing the expression of ACLY and ACC1 [84]. LBX2-AS1 played a critical role in promoting the migration and EMT of ESCC cells, and this role was mediated by HNRNPC through increasing ZEB1 and ZEB2 mRNA stability [68] (Figure 3).

**Figure 3 cancers-14-05139-f003:**
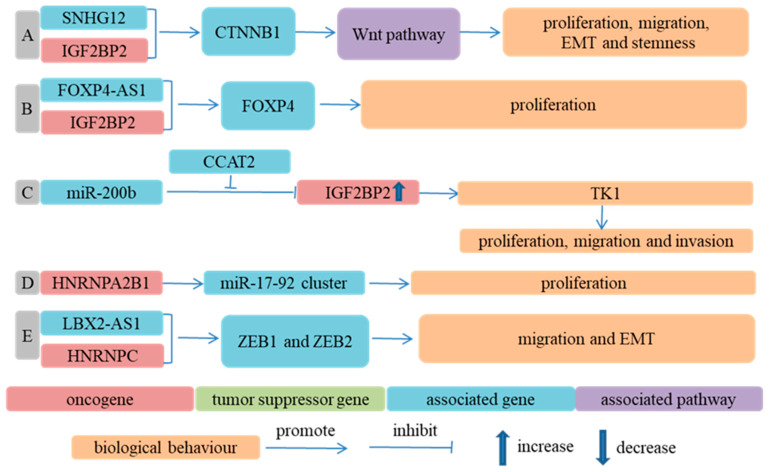
m^6^A RNA binding proteins participate in ESCA progression in an m^6^A-dependent way. (**A**) Ref. [64] IGF2BP2 interacted with SNHG12 to enhance mRNA stability of CTNNB1, elevated expression of β-catenin encoded by CTNNB1, activated Wnt signaling pathway and thus facilitated proliferation, migration, EMT and stemness of ESCC cells; (**B**) Ref. [65] IGF2BP2 interacted with FOXP4-AS1, enhanced FOXP4 mRNA stability, upregulated FOXP4 expression and promoted proliferation of ESCC cells; (**C**) Ref. [66] CCAT2 alleviated inhibition of IGF2BP2 by miR-200b and elevated IGF2BP2 expression, which upregulated expression of TK1 and facilitated proliferation, migration and invasion of ESCC cells; (**D**) Ref. [79] HNRNPA2B1 bound to m^6^A modification sites of miR-17-92 cluster, enhanced expression of miR-17-92 cluster and facilitated proliferation of ESCC cells; (**E**) Ref. [68] Interaction of HNRNPC with LBX2-AS1 enhanced ZEB1 and ZEB2 mRNA stability, upregulated expression of ZEB1 and ZEB2 and then promoted migration and EMT of ESCC cells.

##### m^6^A Regulators Regulated Biological Behavior of ESCC Cells in an m^6^A-Independent Way

Writers

METTL3 was also involved in the progression of ESCC in an m^6^A-independent way [124,126,127]. For example, METTL3 might influence the biological behavior of ESCC cells by regulating Wnt/β-catenin and AKT signaling pathways [124]. In addition, METTL3 facilitated the malignant phenotype of ESCC cells by upregulating the expression of COL12A1 and activating the MAPK signaling pathway [127] (Figure 4).

**Figure 4 cancers-14-05139-f004:**
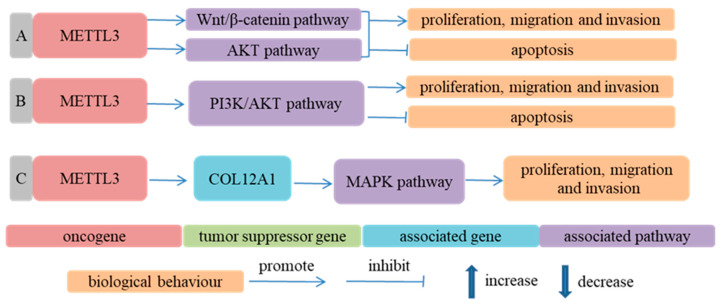
Methyltransferases participate in ESCA progression in an m^6^A-independent way. (**A**) Ref. [124] METTL3 promoted proliferation, migration and invasion of ESCC cells and suppressed apoptosis of ESCC cells by activating Wnt/β-catenin and AKT signaling pathways; (**B**) Ref. [126] METTL3 facilitated proliferation, migration and invasion of ESCC cells and repressed apoptosis of ESCC cells by activating PI3K/AKT signaling pathway; (**C**) Ref. [127] METTL3 increased COL12A1 expression, activated MAPK signaling pathway and promoted proliferation, migration and invasion of ESCC cells.

Erasers

The overexpression of ALKBH5 induced the apoptosis of ESCC cells by changing the expression of Bax, cleaved caspase 3 and Bcl2 in an m^6^A-independent way [86]. FTO promoted the proliferation and migration of ESCC cells by increasing the expression of MMP13 [132] (Figure 5).

**Figure 5 cancers-14-05139-f005:**
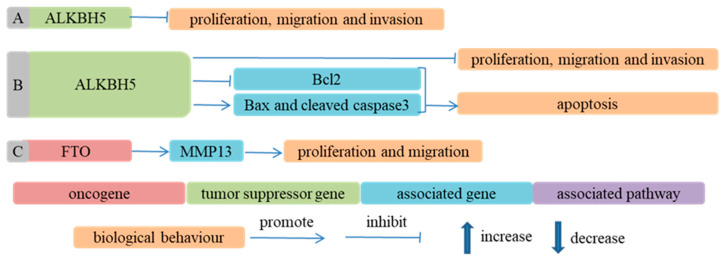
Demethylases participate in ESCA progression in an m^6^A-independent way. (**A**) Ref. [131] ALKBH5 inhibited proliferation, migration and invasion of ESCC cells; (**B**) Ref. [86] ALKBH5 not only inhibited apoptosis of ESCC cells by regulating expression of Bcl2, Bax and cleaved caspase3 but also promoted proliferation, migration and invasion of ESCC cells; (**C**) Ref. [132] FTO facilitated proliferation and migration of ESCC cells through upregulating MMP13 expression.

Readers

The downregulation of YTHDC2 changed the expression of some genes, which were enriched in p53, NF-kappa B and the JAK-STAT signaling pathway. Therefore, the downregulation of YTHDC2 might enhance the growth of ESCC cells through regulating certain pathological signaling pathways [133].

miR-98-5p and miR-454-3p could suppress the biological phenotype of ESCC cells by targeting IGF2BP1 [134,135].

The downregulation of miR-186 alleviated its inhibition of HNRNPC, upregulated the expression of HNRNPC and facilitated the malignant behavior of ESCC cells [136].

EIF3B cooperated with TEX9 to promote the malignant phenotype of ESCC cells via activating the AKT signaling pathway [141]. EIF3B also promoted ESCC cells’ proliferation and invasion and inhibited ESCC cells’ apoptosis through activating the β-catenin signaling pathway [137]. The knockdown of EIF3E repressed the proliferation and migration of ESCC cells [138]. EIF3H could stabilize Snail through deubiquitination and could promote the aggressiveness of ESCC cells [139] (Figure 6).

**Figure 6 cancers-14-05139-f006:**
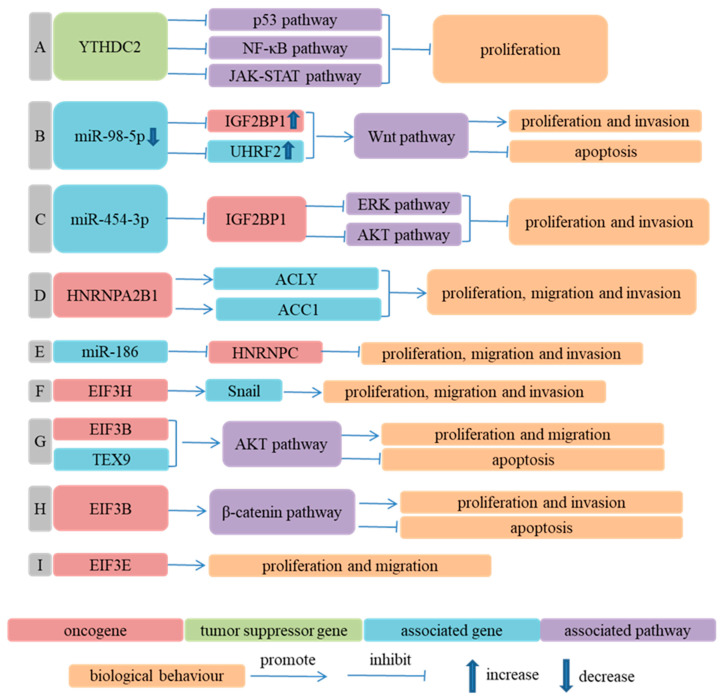
m^6^A RNA binding proteins participate in ESCA progression in an m^6^A-independent way. (**A**) Ref. [133] YTHDC2 inhibited proliferation of ESCC cells through changing expression of some genes enriched in p53 signaling pathway, NF-kappa B signaling pathway and JAK-STAT signaling pathway; (**B**) Ref. [134] IGF2BP1 and UHRF2 interacted to promote proliferation and invasion of ESCC cells and inhibit apoptosis of ESCC cells, which mediated regulation of miR-98-5p on biological phenotype of ESCC cells; (**C**) Ref. [135] miR-454-3p suppressed ERK and AKT signaling pathway by targeting IGF2BP1, then inhibited proliferation and invasion of ESCC cells; (**D**) Ref. [84] HNRNPA2B1 facilitated ESCC progression by accelerating fatty acid synthesis via increasing expression of ACLY and ACC1; (**E**) Ref. [136] miR-186 inhibited malignant behavior of ESCC cells by downregulating expression of HNRNPC; (**F**) Ref. [139] EIF3H promoted the proliferation, migration and invasion of ESCC cells through increasing stability of Snail protein; (**G**) Ref. [141] EIF3B cooperated with TEX9 to promote malignant phenotype of ESCC cells via activating AKT signaling pathway; (**H**) Ref. [137] EIF3B promoted proliferation and invasion of ESCC cells and inhibited apoptosis of ESCC cells through activating β-catenin signaling pathway; (**I**) Ref. [138] Knockdown of EIF3E repressed proliferation and migration of ESCC cells; in other words, EIF3E could facilitate malignant behavior of ESCC cells.

Combined Effects of m^6^A Regulators in Progression of ESCA

METTL3 overexpression enhanced the m^6^A level of adenomatous polyposis coli (APC) mRNA in an METTL14-dependent way, elevated the binding of YTHDF2 with APC mRNA and promoted the degradation of APC mRNA. Ultimately, the downregulation of APC enhanced aerobic glycolysis, ESCC cells growth and tumor formation in mice through activating the Wnt/β-catenin signaling pathway [69].

RBM15 interacted with METTL3 in a WTAP-dependent way to deposit m^6^A onto lncRNA metastasis-associated lung adenocarcinoma transcript 1 (MALAT1), located at nuclear speckles (NSs). YTHDC1 bound to the m^6^A of MALAT1, which maintained the composition of NSs and increased the migration of ESCC cells [82].

The combined regulation of EGR1 by METTL3 and YTHDF3 enhanced the expression of EGR1, activated the Snail signaling pathway and promoted the invasion and metastasis of ESCC cells [72].

Cigarette smoking and sexual hormones might be associated with different incidence rates of ESCC between men and women [145,146,147,148]. Wu et al. [81] reported whether and how a micropeptide affected the progression of male ESCC. The micropeptide was named Yin Yang 1 (YY1)-binding micropeptide (YY1BM), which was encoded by Y-linked lncRNA LINC00278. Specifically, cigarette smoking decreased the methylation of an ALKBH5 promoter, enhanced ALKBH5 expression, downregulated the m^6^A modification of LINC00278 and binding of YTHDF1 with LINC00278 and led to a translation suppression of YY1BM. The downregulation of YY1BM increased the survival of ESCC cells through a series of cascade reactions. Furthermore, Wu et al. [81] also reported that METTL3, METTL14 and WTAP acted as “writers” for the m^6^A modification of LINC00278 to participate in the development of male ESCC.

miR-193a-3p could target ALKBH5 and inhibit the expression of ALKBH5. The knockdown of ALKBH5 increased the m^6^A level of pri-miR-193a-3p and promoted the maturation of pri-miR-193a-3p. Thus, miR-193a-3p and ALKBH5 formed a positive feedback loop, which facilitated the proliferation and metastasis of ESCC cells in vitro and in vivo. The researchers also revealed that METTL3, as a methyltransferase of pri-miR-193a-3p, played an opposite role in the maturation of miR-193a-3p compared to ALKBH5 [80].

The overexpression of FTO promoted the proliferation and cell cycle progression of ESCC cells, which was realized by accelerating the expression of LINC00022 in a YTHDF2-dependent pattern and decay of p21 [83].

FTO could promote the proliferation, migration and invasion of ESCC cells by regulating the m^6^A level and expression of ERBB2 and HSD17B11. Interestingly, YTHDF1, as a reader of ERBB2 and HSD17B11, increased the stability of ERBB2 mRNA but decreased the translation efficiency of HSD17B11 mRNA [73,74].

YTHDF1 could recognize, bind and increase the stability of m^6^A-modified HK2 mRNA and upregulate the expression of HK2, which mediated the role of HCP5 in promoting the progression of ESCC. In this process, METTL3 acted as a methyltransferase of HK2 mRNA [71].

Linc01305 interacted with IGF2BP2 and IGF2BP3 to increase the stability of HTR3A mRNA and expression of HTR3A, and consequently promoted the proliferation and migration of ESCC cells [70] (Figure 7).

**Figure 7 cancers-14-05139-f007:**
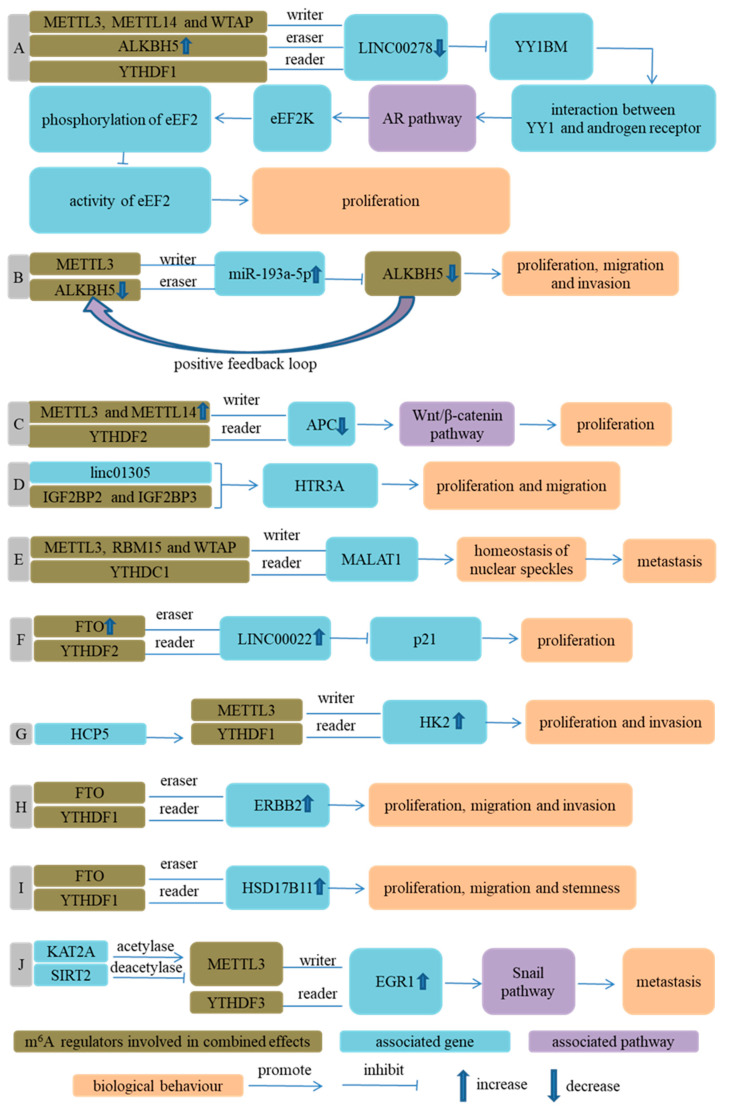
Combined effects of m^6^A regulators on ESCA progression. (**A**) Ref. [81] Cigarette smoking enhanced ALKBH5 expression through decreasing methylation of ALKBH5 promoter, downregulated m^6^A modification of LINC00278 and binding of YTHDF1 with LINC00278 and consequently led to translation suppression of Yin Yang 1 (YY1)-binding micropeptide (YY1BM), a micropeptide encoded by LINC00278. Downregulation of YY1BM promoted the interaction of YY1 with androgen receptor (AR), activated AR signaling pathway and increased eEF2K expression. Under nutrient deprivation, the elevated phosphorylation of eEF2 inhibited activity of eEF2 and translation elongation and increased survival of ESCC cells; (**B**) Ref. [80] miR-193a-3p could target ALKBH5 and inhibit expression of ALKBH5. Interestingly, knockdown of ALKBH5 promoted the maturation of miR-193a-3p. Thus, miR-193a-3p and ALKBH5 formed a positive feedback loop, which facilitated proliferation and metastasis of ESCC cells in vitro and in vivo; (**C**) Ref. [69] Overexpression of METTL3 enhanced m^6^A level of adenomatous polyposis coli (APC) mRNA in an METTL14-dependent way, elevated binding of YTHDF2 with APC mRNA and promoted degradation of APC mRNA. Decreased APC expression facilitated proliferation of ESCC cells through activating Wnt/β-catenin signaling pathway; (**D**) Ref. [70] Linc01305 interacted with IGF2BP2 and IGF2BP3 to increase stability of HTR3A mRNA and expression of HTR3A, then promoted proliferation and migration of ESCC cells; (**E**) Ref. [82] RBM15 interacted with METTL3 in a WTAP-dependent way to deposit m^6^A onto MALAT1, specifically located at nuclear speckles (NSs), and YTHDC1 could bind to m^6^A of MALAT1, which was important for NSs homeostasis and metastasis of ESCC cells; (**F**) Ref. [83] Overexpression of FTO significantly reduced the enrichment of m^6^A at site 2 of LINC00022 transcript and led to a decrease in degradation of LINC00022 by YTHDF2. FTO promoted proliferation of ESCC cells by upregulating expression of LINC00022 in a YTHDF2-dependent pattern; (**G**) Ref. [71] YTHDF1 increased stability of m^6^A-modified HK2 mRNA and upregulated expression of HK2, which mediated the role of HCP5 in promoting progression of ESCC. In this process, METTL3 acted as methyltransferase of HK2 mRNA; (**H**) Ref. [73] FTO promoted proliferation, migration and invasion of ESCC cells by regulating m^6^A level of ERBB2, in which, YTHDF1, as reader of ERBB2, increased stability of ERBB2 mRNA; (**I**) Ref. [74] Combined role of FTO and YTHDF1 resulted in upregulation of HSD17B11 and progression of ESCC, in which, reduced m^6^A level of ERBB2 by FTO led to inhibition of YTHDF1-induced decreased translation efficiency of HSD17B11; (**J**) Ref. [72] Increased m^6^A level of EGR1 mRNA by METTL3 and increased binding of m^6^A-modified EGR1 mRNA with YTDF3 led to upregulation of EGR1, activated Snail signaling pathway and promoted invasion and metastasis of ESCC cells.

Various signaling pathways mediated the regulation of m^6^A regulators on ESCC, consisting of AKT, NF-κB, MAPK, Wnt/β-catenin, etc. Interestingly, the same gene could target different genes or signaling pathways to influence the malignancy of ESCC. For instance, the target genes and signaling pathways of METTL3 included GLS2, NOTCH1, TNFR1, miR-200-5p, miR-320b, miR-99a-5p, Notch pathway, AKT pathway, MAPK pathway, NF-κB pathway and Wnt/β-catenin pathway [59,60,61,75,76,124,126]. Different genes could also regulate the same gene, such as p21, a cell cycle inhibitor, which participated in the regulation process of METTL3, ALKBH5, FTO and YTHDF2 in ESCC [62,83,85]. Different m^6^A-modified genes could play their regulatory role in ESCC through the same pathway, such as the AKT pathway [76,77,124,126,135,141], NF-κB pathway [61,133] and Wnt/β-catenin pathway [64,69,124,134,137]. Therefore, a combined analysis of m^6^A regulators might be beneficial for better understanding their effect and mechanism in ESCA and providing an experimental basis for the drug research of ESCA.

#### 4.2.3. The Role of m^6^A Regulators in Treatment of ESCA

The survival time of ESCA patients has improved dramatically with the improvement in the surgical level and development of drugs. However, the five-year survival rate of ESCA patients remains dismal due to treatment failure resulting from therapeutic resistance. Thus, it is urgent to identify the mechanism responsible for therapeutic resistance and to find new drugs alone or in combination with other therapeutic agents used to improve the prognosis of ESCA patients. Considering the crucial role of m^6^A regulators in the occurrence and progression of ESCA, m^6^A regulators might be promising therapeutic targets. A previous study demonstrated that the depletion of METTL3 enhanced the sensitivity of pancreatic cancer cells to cisplatin and irradiation [149]. Compared to wild-type mice, YTHDF1^-/-^ mice acquired a better therapeutic efficacy of PD-L1 checkpoint blockade, implying the potential of YTHDF1 as a target of antitumor immunotherapy [150]. Similarly, m^6^A regulators played a non-negligible role in the therapeutic resistance of ESCA [67,77,122,129,151].

Platinum-based agents have been widely applied to treat esophageal cancer patients. Platinum treatment resulted in the upregulation of SNHG3. SNHG3 could sponge miR-186-5p, alleviated the inhibition of METTL3 by miR-186-5p, increased the expression of MTLL3 and promoted the growth of ESCC cells in vitro and in vivo. Therefore, targeting SNHG3/miR-186-5p/METTL3 might enhance the platinum efficacy [122]. Hypoxia could induce the expression of some genes associated with resistance to anticancer drugs, then decreased the sensitivity of cells to chemotherapy. Hypoxia induced the expression of lncRNA E2F1 messenger RNA stabilizing factor (EMS), which could sponge miR-758-3p and weaken its repression of WTAP, increased the expression of WTAP and led to resistance of ESCC cells to cisplatin. Therefore, EMS, miR-758-3p and WTAP might be considered as a therapeutic target to enhance cisplatin efficiency [129].

m^6^A regulators were also associated with radioresistance. Cancer stem-like cells (CSCs) have been reported to be associated with the radioresistance of ESCC cells [152]. METTL14 was downregulated in CSCs, which resulted in a decreased expression of miR-99a-5p in an m^6^A-dependent manner. The downregulation of miR-99a-5p led to an increase in the expression of TRIB2. TIRB2 bridged METTL14 to COP1 and one E3 ligase, and resulted in the ubiquitination and degradation of METTL14. Thus, METTL14, miR-99a-5p and TRIB2 formed a positive feedback circuit. TRIB2 could promote CSCs persistence and the radioresistance of ESCC cells through activating the Akt/mTOR/S6K1 signaling pathway [77]. IGF2BP3 enhanced KIF18A mRNA stability, upregulated KIF18A expression and facilitated ESCC cells’ proliferation, migration, invasion and radioresistance [67]. The knockdown of IGF2BP3 increased the susceptibility of radioresistant TE-5 and TE-9 cells to radiotherapy [153]. These results suggested that METTL14/miR-99a-5p/TRIB2 axis and IGF2BP3 might be potential targets for improving the sensitivity of ESCC cells to radiotherapy.

m^6^A regulators could be used as a treatment target of herbal medicine for esophageal cancer. HNRNPA2B1 codes for two isoforms, HNRNPA2 and HNRNPB1. The crude extract of a South African medicinal plant, Cotyledon orbiculata, reduced colon cancer cells and ESCC cells proliferation and induced their apoptosis through regulating the alternative splicing of HNRNPA2B1 (decreasing the expression of HNRNPB1). A further investigation on HCT116 cells showed that the silencing of HNRNPB1 led to a switch in the splicing of BCL2L1 from a Bcl-xL anti-apoptotic isoform to a Bcl-xS pro-apoptotic isoform and promoted the apoptosis of HCT116 cells. Thus, HNRNPA2B1 might be a potential treatment target of herbal medicine for ESCA. [151].

Elvitegravir, originally developed to treat human immunodeficiency virus (HIV) infection, could repress the invasion and metastasis of ESCC cells by enhancing the proteasomal degradation of METTL3 mediated by STUB1 [72]. If the therapeutic effect of elvitegravir is proven clinically, there will be an additional option for the treatment of ESCA.

#### 4.2.4. The Association between Expression of m^6^A Regulators and Prognosis of ESCA Patients

The association between the expression of m^6^A regulators and prognosis of ESCA patients was shown in Appendix A. A higher expression or positive expression of METTL3, WTAP, HNRNPA2B1, HNRNPC and EIF3B was associated with a poor prognosis of ESCA patients [59,60,61,69,72,79,123,126,127,129,130,137]. On the contrary, a lower METTL14 expression predicted a poor overall survival (OS) of ESCA patients [77]. There was no relation between the expression of EIF3E and prognosis of ESCA patients [138]. Xu et al. [130] revealed that a low ALKBH5 expression was associated with a poor survival of ESCA patients. Similarly, ESCA patients with a positive ALKBH5 expression showed a longer OS and disease-free survival (DFS) and less recurrence/distant metastasis after surgery than those with a negative ALKBH5 expression [78]. However, a study on 177 patients showed that a higher ALKBH5 expression was related to a lower OS [62]. There were two studies that reported the association of FTO expression with the prognosis of ESCA patients. One showed the relation of a poor OS with a high FTO expression [74]; the other suggested no relation between them [62]. The difference in subjects, sample size and detection methods might partly explain the discrepancy. For patients treated with surgery alone, a high IGF2BP3 expression was associated with a poor OS, disease-specific survival (DSS) and DFS [154]. Takata et al. [140] reported that IGF2BP3-positive patients had a poorer OS and recurrence-free survival (RFS) than IGF2BP3-negative patients.

Present studies on the association between METTL3 expression and the prognosis of ESCA patients reached a consistent conclusion, which was that a “higher METTL3 expression predicted a poor prognosis of ESCA patients”. Results about ALKBH5 and FTO were opposite or inconsistent. The number of studies about other m^6^A regulators is limited. Therefore, further studies are needed to comprehensively analyze the relation between the expression of m^6^A regulators and the survival of ESCA patients due to the complexity of the m^6^A regulatory network. Based on information from the public database, some researchers constructed a prognostic signature according to the expression of m^6^A regulators to predict the survival of ESCA patients [79,84,130,136,155,156]. For example, Xu et al. [130] analyzed the RNA sequencing transcriptome data of 13 m^6^A regulators from the TCGA ESCA database, constructed a prognostic signature, which consisted of HNRNPC and ALKBH5, and established a formula to calculate the risk score. According to the risk score, the ESCA patients were divided into a low risk group and high risk group. Compared with the low risk group patients, the patients in the high risk group had a significantly poorer survival. Therefore, the prognostic signature could be used as an independent prognostic marker for ESCA patients.

## 5. Conclusions and Perspectives

After transcription, different chemical modifications can occur on cellular RNA, such as N6-methyladenosine (m^6^A), N6,2′-O-dimethyladenosine (m^6^Am), N1-methyla- denosine (m^1^A) and 5-methylcytosine (m^5^C) [157]. Among them, m^6^A is the most abundant RNA modification, and is dynamic and reversible. m^6^A is catalyzed by methyltransferase, then removed by demethylase or recognized by m^6^A RNA binding protein. During the development and progression of ESCA, m^6^A on various types of RNA was involved in all steps of RNA metabolism, post-transcriptionally regulated the expression of the gene and affected the biological behavior of ESCC cells. m^6^A regulators not only regulated the expression of various type of RNAs but also received regulation by genetic variation, epigenetic modification and non-coding RNAs. During the development of ESCA, METTL3 acted as an oncogene. The mRNA expression level of METTL3 might be used for the diagnosis of ESCA with a sensitivity of 75% and specificity of 72.06%. Furthermore, a high SUVmax of 18F-FDG PET/CT could be used as a non-invasive predictive marker for a high METTL3 expression of ESCA patients. m^6^A regulators participated in the progression of ESCA in an m^6^A-dependent or m^6^A-indepent manner. The same m^6^A regulator could target different genes or signaling pathways to influence the malignancy of ESCA. Similarly, different m^6^A-modified genes could play their regulatory role in ESCA through the same gene or signaling pathway. The m^6^A regulators played a key role in the occurrence and progression of ESCA, which suggested that they might serve as potential therapeutic targets for ESCA. At present, chemotherapy and radiotherapy remain the primary treatment for ESCA. Platinum resistance and radioresistance often result in a poor prognosis of ESCA patients. Targeting the SNHG3/miR-186-5p/METTL3 axis and EMS/miR-758-3p/WTAP axis might help to increase the sensitivity of ESCC cells to platinum; the METTL14/miR-99a-5p/TRIB2 axis and IGF2BP3 might be useful targets for decreasing the radioresistance of ESCC cells. In addition, an extract of Cotyledon orbiculata, a South African medicinal plant, exerted an anti-proliferative effect in ESCC cells through possibly regulating the alternative splicing of hnRNPA2B1 [151]. Moreover, elvitegravir could enhance the proteasomal degradation of METTL3 mediated by STUB1 to repress the invasion and metastasis of ESCC cells [72]. A prognostic signature based on m^6^A regulators is helpful for predicting the prognosis of ESCA patients accurately.

Of note, YTHDF1 has been widely reported to promote protein synthesis. However, YTHDF1 decreased the translation efficiency of HSD17B11 mRNA in the progression of ESCC. In addition, whether ALKBH5 and FTO play roles as an oncogene or cancer suppressor gene is inconclusive, which might be associated with a difference in the sample size, detection method and cell lines employed. In future, a combined analysis of m^6^A regulators in a multi-center study with a large sample size and identical detection method might be beneficial for better understanding their effect and mechanism in ESCA and for drawing a consistent and reliable conclusion.

Inflammation is a physiological response of organisms to various injuries, such as an infection of microorganisms, autoimmunity and physical damage. The disequilibrium of pro-inflammatory factors and anti-inflammatory factors will result in a persistent inflammatory state, so-called chronic inflammation, in which, a large number of cytokines arise and facilitate the initiation, promotion, invasion and metastasis of a tumor [158]. Previous studies revealed a close relation between inflammatory esophageal disease and ESCA [159,160]. In particular, in 58 ESCA patients and 10,614 non-ESCA subjects who received endoscopic examination, there was a significantly higher proportion of esophagitis in ESCA patients (89.7%) than in non-ESCA subjects (14.3%) [161]. The association of m^6^A regulators with pro-inflammatory genes in tumor development and progression has been widely reported. YTHDF2 deletion aggravated the inflammation, angiogenesis and metastatic progression of hepatocellular carcinoma [55]. In another study on intrahepatic cholangiocarcinoma, FTO changed the expression of inflammatory genes via upregulating NR5A2 [162]. In ESCA, there is no similar report. Identifying the target of m^6^A regulators during the process from chronic esophagitis to ESCA would help to determine the pathogenesis of ESCA and provide experimental evidence for the development of a new drug.

To meet the needs of uncontrolled proliferation, one hallmark of cancer, cancer cells must acquire sufficient nutrients from the surrounding tumor microenvironment through regulating various metabolism-associated enzymes, signaling pathways and transcription factors [163]. This ability of cancer cells is termed metabolic reprogramming, which includes the metabolic recombination of glucose, lipid and glutamine; for instance, whether under aerobic conditions or under anoxic conditions cancer cells depend on glycolysis to obtain energy preferentially, which is also known as the “Warburg effect” [164,165]. Glucose transporter 1 (GLUT1) and hexokinase 2 (HK2) are rate-limiting enzymes of the “Warburg effect”, responsible for glucose transport and transformation, respectively. Based on multiple public databases, some studies showed a relation between the expression of these two enzymes and m^6^A regulators, which hinted that m^6^A-modified genes participated in glucose metabolic reprogramming [66,125,166], which was validated in some cytological experiments [69,71]. Pyruvate kinase M2 isozyme (PKM2), as a downstream gene of APC regulated by METTl3, METTL14 and YTHDF2, enhanced aerobic glycolysis and, in turn, promoted the proliferation of ESCC cells [69]. The combined effect of METTL3 and YTHDF1, as a writer of HK2 mRNA and reader of m^6^A-modified HK2 mRNA, respectively, caused an upregulation of HK2 and promoted the glycolysis of ESCC cells [71]. m^6^A regulators were also involved in lipid metabolic reprogramming [74,84]. The knockdown of HNRNPA2B1 inhibited the malignant biological behavior of ESCC cells by downregulating the expression of de novo fatty acid synthetic enzymes and suppressing lipid accumulation in ESCC cells [84]. FTO facilitated the growth and invasion of ESCC cells by enhancing the formation of lipid droplets via the regulation of HSD17B11 expression in ESCC cells [74]. So far, whether m^6^A regulators affect glutamine metabolic reprogramming in ESCA has not been disclosed. Targeting metabolism-associated m^6^A regulators might be new strategies for the treatment of ESCA. As shown in one study, some drugs targeting glutaminase isoenzymes had an anticancer effect [167]. Determining the role of m^6^A regulators in ESCA metabolic reprogramming will be beneficial for the development of a new drug.

Immune escape is one of the basic characteristics of tumor cells. The application of immune checkpoint inhibitors (ICIs) that activate antitumor immune cells has shown a significant therapeutic effect on various malignant tumors, including ESCA [168]. However, only a few patients benefit from ICIs treatment because of resistance, thus limiting its application. The tumor-immune microenvironment (TIME), consisting of PD-L1 expression on tumors, tumor-infiltrating lymphocytes, tumor-associated macrophages, etc., might impact the patients’ response to ICIs. m^6^A modification is involved in remodeling TIME through regulating the expression of PD-L1 on tumors and regulating the differentiation and activation of tumor-associated immune cells [169,170,171,172,173]. The continual expression of PD-L1 in tumor cells facilitates tumor immune escape. IGF2BP1 recognized PD-L1 mRNA modified by METTL3 and enhanced the expression of PD-L1 in bladder cancer cells, which decreased the attack of CD8^+^ T cells [170]. The deletion of METTL3 disturbed T cells differentiation and macrophage activation via regulating different downstream target genes [172,173]. In addition, a change in the expression of m^6^A regulators also might influence the distribution and function of immune cells in TIME [174,175,176]. In hepatocellular carcinoma, ALKBH5 promoted macrophage recruitment through upregulating MAP3K8 expression in an m^6^A-dependent manner [174]. In ESCA, some studies revealed that the expression of m^6^A regulators was associated with PD-L1 expression and immune cells infiltration, suggesting the role of m^6^A regulators as an immune therapy target individually or in combination with ICIs [128,155,156,177]. However, how m^6^A regulators affect the TIME of ESCA has not been reported, which provides a direction for future research in ESCA.

In summary, determining the role and mechanism of m^6^A modification and m^6^A regulators in the occurrence, progression, remedy and prognosis of ESCA is necessary for an early diagnosis, individualized treatment and improved prognosis of ESCA patients. Identifying the target of m^6^A regulators during the process from chronic esophagitis to ESCA, the role of m^6^A regulators in metabolic reprogramming and the impact of m^6^A regulators on TIME in ESCA might be a future research direction.

## Figures and Tables

**Table 1 cancers-14-05139-t001:** The m^6^A methylation modified various RNAs in ESCA.

m^6^A Regulator	Target	Type of Target RNA	Function	Molecular Mechanism	Reference
METTL3	GLS2	mRNA	Writer	Increase m^6^A level of GLS2 mRNA, upregulate GLS2 expression	[59]
METTL3	NOTCH1	mRNA	Writer	Increase m^6^A level of NOTCH1 mRNA, upregulate NOTCH1 expression	[60]
METTL3	TNFR1	mRNA	Writer	Increase m^6^A level of TNFR1 mRNA, upregulate TNFR1 expression	[61]
ALKBH5	CDKN1A	mRNA	Eraser	Decrease m^6^A level of CDKN1A mRNA, decrease stability of CDKN1A mRNA, downregulate p21 expression	[62]
FTO	SIM2	mRNA	Eraser	Decrease m^6^A level of SIM2 mRNA, decrease stability of SIM2 mRNA, downregulate SIM2 expression	[63]
IGF2BP2	CTNNB1	mRNA	Reader	Increase stability of CTNNB1 mRNA, upregulate CTNNB1 expression	[64]
IGF2BP2	FOXP4	mRNA	Reader	Increase stability of FOXP4 mRNA, upregulate FOXP4 expression	[65]
IGF2BP2	TK1	mRNA	Reader	Recognize m^6^A of TK1 mRNA, upregulate TK1 expression	[66]
IGF2BP2	KIF18A	mRNA	Reader	Increase stability of KIF18A mRNA, upregulate expression of KIF18A	[67]
HNRNPC	ZEB1 and ZEB2	mRNA	Reader	Increase stability of ZEB1 and ZEB2 mRNA, upregulate expression of ZEB1 and ZEB2	[68]
METTL3, METTL14, and YTHDF2	APC	mRNA	METTL3 and METTL14 as writer; YTHDF2 as reader	METTL3 increases m^6^A level of APC mRNA in an METTL14-dependent way; YTHDF2 promotes degradation of APC mRNA	[69]
IGF2BP2 and IGF2BP3	HTR3A	mRNA	Reader	Increase stability of HTR3A mRNA, upregulate HTR3A expression	[70]
METTL3 and YTHDF1	HK2	mRNA	METTL3 as writer; YTHDF1 as reader	Increase stability of HK2 mRNA, upregulate HK2 expression	[71]
METTL3 and YTHDF3	EGR1	mRNA	METTL3 as writer; YTHDF3 as reader	Increase stability of EGR1 mRNA, upregulate EGR1 expression	[72]
FTO and YTHDF1	ERBB2	mRNA	FTO as eraser; YTHDF1 as reader	Increase stability of ERBB2 mRNA, upregulate ERBB2 expression	[73]
FTO and YTHDF1	HSD17B11	mRNA	FTO as eraser; YTHDF1 as reader	Decrease the translation efficiency of HSD17B11 mRNA, downregulate HSD17B11 expression	[74]
METTL3	pri-miR-200-5p	miRNA	Writer	Increase m^6^A level of pri-miR-200-5p, upregulate miR-200-5p expression	[75]
METTL3	pri-miR-320b	miRNA	Writer	Increase m^6^A level of pri-miR-320b, upregulate miR-320b expression	[76]
METTL14	pri-miR-99a	miRNA	Writer	Increase m^6^A level of pri-miR-99a, upregulate miR-99a-5p expression	[77]
ALKBH5	pri-miR-194-2	miRNA	Eraser	Decrease m^6^A level of pri-miR-194-2, downregulate miR-194-2 expression	[78]
HNRNPA2B1	miR-17-92 cluster	miRNA	Reader	Bind to m^6^A of miR-17-92 cluster, upregulate expression of miR-17-92 cluster	[79]
METTL3 and ALKBH5	pri-miR-193a-3p	miRNA	METTL3 as writer; ALKBH5 as eraser	METTL3 increases m^6^A level of pri-miR-193a-3p, upregulates miR-193a-3p expression; ALKBH5 decreases m^6^A level of pri-miR-193a-3p, downregulates miR-193a-3p expression	[80]
METTL3, METTL14, WTAP, ALKBH5, and YTHDF1	LINC00278	lncRNA	METTL3, METTL14, and WTAP as writers; ALKBH5 as eraser; YTHDF1 as reader	METTL3, METTL14, and WTAP increase m^6^A level of LINC00278; ALKBH5 decreases m^6^A level of LINC00278; YTHDF1 promotes translation of LINC00278	[81]
METTL3, RBM15, WTAP, and YTHDC2	MALAT1	lncRNA	METTL3, RBM15, and WTAP as writer; YTHDC2 as reader	RBM15 interacts with METTL3 in a WTAP-dependent way to deposit m^6^A onto MALAT1; YTHDC1 binds to m^6^A of MALAT1 and maintains composition of nuclear speckle	[82]
FTO and YTHDF2	LINC00022	lncRNA	FTO as eraser; YTHDF2 as reader	FTO reduces the enrichment of m^6^A at site 2 of LINC00022 transcript; YTHDF2 promotes degradation of LINC00022	[83]

**Table 2 cancers-14-05139-t002:** The m^6^A regulators regulated by non-coding RNAs in ESCA.

m^6^A Regulator	Upstream Gene	Type of Upstream Gene	Molecular Mechanism	Reference
METTL3	SNHG3 and miR-186-5p	lncRNA and miRNA	SNHG3 sponges miR-186-5p and alleviates inhibition of METTL3 by miR-186-5p	[122]
WTAP	EMS and miR-758-3p	lncRNA and miRNA	EMS sponges miR-758-3p and alleviates inhibition of METTL3 by miR-758-3p	[129]
IGF2BP2	CCAT2 and miR-200b	lncRNA and miRNA	CCAT2 sponges miR-200b and alleviates inhibition of IGF2BP2 by miR-200b	[66]
IGF2BP1	miR-454-3p	miRNA	miR-454-3p inhibits expression of IGF2BP1	[135]
HNRNPC	miR-186	miRNA	miR-186 inhibits expression of HNRNPC	[136]
ALKBH5	miR-193a-3p	miRNA	miR-193a-3p inhibits expression of ALKBH5	[80]

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
