# Peer review of "The Role of m6A Modification and m6A Regulators in Esophageal Cancer"

_cancers, 2022, doi:10.3390/cancers14205139_

Round 1

Reviewer 1 Report

Li et al review titled "The role of m6A modification and m6A regulators in esophageal cancer ", describing the role of relatively newly discovered m6A modification and associated readers, writers and erasers in initiation and progression of esophageal cancer is comprehensive and thorough. Considering that it is relatively new area of research and interest, majority of the review described unique studies not thorough concepts specific to esophageal cancer. It would be helpful to readers, if authors can tabulate the mechanisms described in figures.

Author Response

Thank you for your comments and suggestions. I understand the reviewer's consideration. However, I privately think the figures and their annotations can help readers understand the role and mechanism of m6A regulators in the progression and treatment of esophageal cancer quickly. Moreover, the text already contains a large number of figures and tables. I'm not sure whether my opinion is appropriate. If it is necessary to tabulate the mechanisms described in figures, I sincerely request the reviewer to provide the authors with guidance and assistance for specific methods. Then, I can make corresponding modifications. Thanks for your review and suggestions again.

Reviewer 2 Report

The manuscript is well written and given a very extensive and detailed overview of pretty much all things related, maybe related and somewhat possibly related to m6A modification in esophageal cancer. The manuscript would benefit from extensive shortening and focusing on only the truly associated genes and pathways and only those related to esophageal cancer. Furthermore, a summary of tables could be depicted, moving the very lengthy tables to the supplementary. There is so much information in it, the reader (or at least me) loses focus on hat is important, the message in the table.

Author Response

Thank you for your comments and suggestions. The manuscript analyzed the role and possible mechanism of m6A regulators in the occurrence, progression, treatment and prognosis of esophageal cancer comprehensively. All the m6A regulators played critical roles in esophageal cancer and were related to esophageal closely. Indeed, there is a certain degree of repetition between the content of the text and the content of Figure 8 in “4.2.3. The role of m6A regulators in treatment of ESCC”. Therefore, this section was rewritten and Figure 8 was deleted. The tables showed the results of relevant researches in detail, which led to too large tables and less prominent emphasis. Therefore, table 2 and table 4 were moved to the supplementary material. Thanks for your review and suggestions again.

Round 2

Reviewer 2 Report

authors have substantially shortened and improved the readability of the manuscript. One small remarks. It would beneifit the reader (and scope) to define m6A regulators in the beginning of the manuscript.

Author Response

Thanks for your comments and recommendations. According to your suggestions, we have made corresponding modifications. Thank you again.